# Analysis of the Maximum Efficiency and the Maximum Net Power as Objective Functions for Organic Rankine Cycles Optimization

**DOI:** 10.3390/e25060882

**Published:** 2023-05-31

**Authors:** Johan González, José Matías Garrido, Héctor Quinteros-Lama

**Affiliations:** 1Departamento de Tecnologías Industriales, Faculty of Engineering, Universidad de Talca, Merced 437, Curicó 3340000, Chile; 2Departamento de Ingeniería Química, Faculty of Engineering, Universidad de Concepción, Concepción 4030000, Chile; josemagarrido@udec.cl

**Keywords:** organic Rankine cycle (ORC), maximum efficiency, maximum net power output, PC-SAFT, fluorocarbons, hydrocarbons

## Abstract

Maximum efficiency and maximum net power output are some of the most important goals to reach the optimal conditions of organic Rankine cycles. This work compares two objective functions, the maximum efficiency function, β, and the maximum net power output function, ω. The van der Waals and PC-SAFT equations of state are used to calculate the qualitative and quantitative behavior, respectively. The analysis is performed for a set of eight working fluids, considering hydrocarbons and fourth-generation refrigerants. The results show that the two objective functions and the maximum entropy point are excellent references for describing the optimal organic Rankine cycle conditions. These references enable attaining a zone where the optimal operating conditions of an organic Rankine cycle can be found for any working fluid. This zone corresponds to a temperature range determined by the boiler outlet temperature obtained by the maximum efficiency function, maximum net power output function, and maximum entropy point. This zone is named the optimal temperature range of the boiler in this work.

## 1. Introduction

The study and analysis for improving and optimizing renewable energy sources have increased exponentially in the last few years. The main goal of these actions is to minimize the effect of the greenhouse gases (GHS) produced by conventional energy sources on the generation of electricity, such as fossil fuels combustion (petroleum, carbon, natural gas) [1]. In this regard, some of the most promising renewable energies are solar, wind, hydro, geothermal, and residual energy [2,3,4].

Each one of these renewable energies depends on geographic characteristics. For instance, two-thirds of the energy production in the Nordic countries is produced by renewable energies. Different biomass sources are burned in combined heat and power plants across Finland and Sweden, while Denmark has the highest share of wind power worldwide. Additionally, it is well-known that Iceland generates significant electricity from geothermal sources [5]. However, the world has countries with the potential to massively expand renewable energy sources [6].

The organic Rankine cycles (ORCs) are one of the most popular technologies for renewable energy exploitation, such as geothermal, residual, and solar thermal energy [7,8]. Several criteria have been used to obtain the optimal operational conditions for an ORC. Some of these criteria are the ORC layout [8,9], working fluids selection [10,11], working fluid mixture [2,12,13], and energy source [14,15,16]. In order to select the best operational variables to obtain the maximum performance of the ORC, different optimization approaches are used. The most frequently used objective functions for analyzing and optimizing an ORC are the search for the maximum efficiency and the maximum net power output [8,14,16,17,18,19,20,21].

Aljundi [20] uses the efficiency calculated by the first and second laws as the objective functions to determine the optimal conditions of dry hydrocarbons. The performance of some of the dry hydrocarbons, such as *n*-hexane, isopentane, and *n*-butane, outperforms conventional refrigerants. Wang et al. [19] use an objective function that combines the efficiency and the net power output to select the optimal working fluid as a function of the temperature source, where R123 is the best choice for temperatures ranging between 100 and 180 °C. He et al. [21] employ both the net power output and thermal efficiency in terms of latent heat and the specific heat as objective functions. The results show that the liquid-phase heat capacity and heat of evaporation play a leading role in selecting the working fluids and thermal energy source. Kong et al. [16] utilize thermal and exergetic efficiency as objective functions to study an ORC with R245fa for several sources. The results show that the combined hot water/saturated vapor generates the lowest exergy destruction in the ORC evaporator and provides the highest value of exergetic efficiency.

In the same spirit, He et al. [21] and Zhang et al. [18] employed the net power output and thermal efficiency as objective functions to select the optimal working fluid depending on the type of heat source. The work concludes that dodecane can be a good choice for open-type heat sources. Furthermore, the R123 is an acceptable alternative for utilizing closed-type heat sources. Subsequently, Zhang et al. [17] determine the most unfavorable condensator temperature for common working fluids by a net power optimization. Maali et al. [8] analyze solar and geothermally activated ORC by both traditional optimization functions. The study determines that a heat recovery layout is the most efficient configuration for these applications. However, its efficiency and net power output vary depending on the year station. Goyal et al. [14] used thermal efficiency as an objective function to develop a second-order non-linear polynomial mathematical model. The results indicate that the thermal efficiency is mainly affected by the heat sink temperature, followed by turbine isentropic performance and mass fraction of zeotropic mixture.

This work develops an accurate mathematical structure to determine the maximum net power of a single-stage ORC. The cited structure is compared with the maximum efficiency function [22] and the geometry of the saturated entropy curve [23]. The comparison sheds light on the correct methodologies to obtain the best ORC performance in different circumstances of operation. The rest of the work is organized into a methodology section, qualitative and quantitative analyses, and finalizing with concluding remarks.

## 2. Materials and Methods

### 2.1. Modelling

In order to compare the behavior of both objective functions, a quantitative and qualitative equation of state have been used. On the one hand, the van der Waals EOS [24] is a simple model to study the complex behavior of the working fluids [25]. The above EOS can be written as
(1)$$\frac{{\tilde{A}}^{r}}{RT}=\ln\frac{\tilde{v}}{\tilde{v}-b}-\frac{a}{\tilde{v}RT}$$
where ${\tilde{A}}^{r}$ is the residual Helmholtz energy function, $\tilde{v}$ is the molar volume, *R* is the universal gas constant, *T* the absolute thermodynamic temperature, and *a* and *b* stand as the dispersive constant and the *covolume*, respectively.

On the other hand, the SAFT-family equations of state (EOS) [26,27,28] are molecular-based models on statistical thermodynamics principles [29,30,31]. Particularly, PC-SAFT is an accurate equation of state for predicting the thermophysical properties of complex working fluids [32,33]. PC-SAFT EOS validates its accuracy for its good results and predictive capabilities in a wide range of temperatures and pressure for pure and mixture working fluids [3,13,34,35,36,37,38].

PC-SAFT EOS is expressed as a sum of different contributions to the residual Helmholtz energy of the system, ${\tilde{A}}^{r}$. For a non-associative and non-polar compound, the construction above yield is
(2)$${\tilde{A}}^{r}={\tilde{A}}^{\mathrm{hc}}+{\tilde{A}}^{\mathrm{disp}}$$

In Equation (Equation 2), the superscripts “hc” and “disp” concern the hard-chain and the dispersive contributions, respectively. The details of the EOS model can be found in the original source [32].

The total Helmholtz energy function is the sum of the residual contribution and the ideal contribution, i.e., $\tilde{A}={\tilde{A}}^{r}+{\tilde{A}}^{i}$, where the ideal contribution is given by
(3)$$\frac{{\tilde{A}}^{i}}{RT}=-\ln\tilde{v}-\frac{1}{RT}\;\int \int \;\frac{C_{P}^{i}-R}{T}dT^{2}+\ln\frac{RT^{\Theta }}{P^{\Theta }}+{\tilde{A}}^{i\Theta }$$

In Equation (Equation 3), $T^{\Theta }$ and $P^{\Theta }$ are the temperature and pressure of a reference state Θ, respectively. Meanwhile, ${\tilde{A}}^{i\Theta }$ corresponds to the value of the Helmholtz energy function of the perfect gas at that reference state and $C_{P}$ is the isobaric heat capacity at the perfect gas state. The heat capacity is usually fitted in a polynomial form as
(4)$$\frac{C_{P}^{i}}{R}=\alpha_{0}+\alpha_{1}T+\alpha_{2}T^{2}+\alpha_{3}T^{-2}$$
where *T* is the thermodynamic temperature, $C_{P}^{i}$ is the isobaric heat capacity, and $\alpha_{i}$ are the fitted parameters listed in Table 1. Additionally, Table 2 shows the PC-SAFT parameters for the analyzed working fluids.

### 2.2. Efficiency and Net Power Output as an Objective Function

The single-stage ORC comprises a condenser, a pump, a boiler or evaporator, and a vapor turbine. As shown in Figure 1, a saturated liquid at point (1) is compressed to the boiler pressure at point (2). From point (2), the fluid is isobarically heated, reaching the saturated vapor state at point (3). The saturated vapor is then isentropically expanded, reaching the pressure of the condenser (4), and then it is brought to saturated liquid.

The efficiency of a single-stage ORC is given by the ratio between net power output, $W_{n}$, and the heat used in the boiler, $Q_{b}$. The above yields
(5)$$\eta =\frac{W_{n}}{Q_{b}}$$
where $W_{n}$ can be written as $Q_{b}-Q_{c}$, $Q_{c}$ being the amount of heat exacted in the condenser.

Numerically, the maximum efficiency limit of the working fluid can be obtained through the derivative of Equation (5), which is expanded as
(6)$$\frac{d\eta }{dT_{3}}=\frac{1}{Q_{b}}\left(\frac{dW_{n}}{dT_{3}}-\eta \frac{dQ_{b}}{dT_{3}}\right)$$

Therefore, the optimal operation point of an ORC is given by the function of the maximum efficiency as
(7)$$\eta_{\mathrm{op}}=\frac{dW_{n}}{dT_{3}}{\left(\frac{dQ_{b}}{dT_{3}}\right)}^{-1}$$

Equation (Equation 7) can be applied in two scenarios: firstly, to obtain the maximum cycle efficiency under a fixed operation condition of the condenser. In this case, depending on the temperature and the entropy geometry, the expansion in the turbomachinery can yield a dry or partially wet fluid expansion. The second scenario imposes the condition of a completely dry expansion after the turbine, where the limiting efficiency of the fluids is obtained. In this case, no other combination of temperature and pressures can provide a larger efficiency without regard to heat and head losses [22].

The ψ-function relates to the geometry of the entropy. The line represents the set of extreme values of the saturated vapor line or equivalently
(8)$$\frac{d\tilde{S}}{dT}=-\frac{d^{2}\tilde{G}}{dT^{2}}-\frac{d^{2}\tilde{G}}{dTdP}\frac{dP}{dT}=0$$

Therefore, the ψ-function sheds light on the geometry and transitions between a wet and a dry fluid. This structure has a configurational nature. Hence, it is independent of the thermal contribution of the entropy [23]. For further details, the Appendix A explain the optimization function β and the geometric function ψ.

However, the most usual way to optimize ORC is to maximize the net power output [8,9,18,19,21,41]. The net power output maximum can be obtained by solving the derivative of the expander enthalpy change. The latter is given by
(9)$$\frac{dW_{n}}{dT_{3}}=\frac{d{\tilde{H}}_{3}}{dT_{3}}-\frac{d{\tilde{H}}_{2}}{dT_{3}}-\frac{d{\tilde{H}}_{4}}{dT_{3}}+\frac{d{\tilde{H}}_{1}}{dT_{3}}=0$$

The problem of analytically obtaining and expressing the optimum operation of both the maximum efficiency and the maximum net power output of an ORC is reduced to describe the derivatives of each phase in operation. The details of the derivatives and their application of a generic equation of state are described in previous work [22]. The systematic comparison of both approaches yields a suitable range of pressure for the boiler. Despite the above, the approach can also be applied to other fixed variables. For instance, using a fixed condenser temperature, the boiler temperature can be obtained, which allows to reach the system’s maximum efficiency. Notably, net power output cannot match the optimal efficiency at the same condensation temperature or pressure. The aforementioned fact can be observed in Equations (Equation 7) and (Equation 9). The cancelation of Equation (Equation 9) results in a null value for Equation (Equation 7), considering that the optimum point of efficiency is a single value. The same is valid for limiting fluid and optimal efficiency at a fixed condenser value.

Figure 2 shows three functions that describe different types of transitions for vdW working fluids. The ψ-function is a geometric representation of the entropy in the projection temperature vs. entropy [23]. The β-function represents the condenser and boiler temperature values for an ORC expanding from a saturated vapor from the boiler outlet to a saturated vapor in the turbine outlet. The above function represents an optimal condition of an ORC that reaches the global limit of the efficiency of a working fluid [22]. Similar to the β-function, the ω-function represents the temperature values of the condenser and boiler for an ORC that expands from a saturated vapor from the boiler outlet until a saturated vapor in the turbine outlet with maximum net power output, determining an optimal condition of an ORC that reaches the limit of the net power output of a working fluid.

The ω-function always lies under the line of the optimal limiting efficiency. Equivalently, the present function has a larger range of temperatures in which the outlet temperature of the turbine corresponds to an overheated vapor. A side effect of the above is the condenser’s larger cooling capacity requirement compared with the parameters obtained by reaching optimal efficiency. Furthermore, lower boiler temperatures are needed to reach the optimal net power compared to the optimal efficiency of the cycle. At a simple glimpse, it is noted that the difference between both optimization methods is reduced as the working fluid becomes drier.

For example, taking Figure 2 as a reference, a vdW fluid with $C_{P}/R=7.6$ is a dry fluid for cutting ψ-function in two points. The maximum saturated entropy lies in Point (A), while Point (B) is its extreme value at low temperatures. The limiting efficiency of this fluid is reached when the ORC works between Point (C) and Point (D). As a consequence, at the limiting efficiency, the outlet of the turbine is always partially condensed. The maximum net power is produced when the boiler produces saturated vapor at Point (F), and the fluid is condensed at Point (E).

## 3. Efficiency and Net Power Output for vdW Working Fluids

Figure 3 shows the comparison of efficiency and the net power output obtained both by the $\eta_{\mathrm{op}}$ and by the $W_{\mathrm{op}}$. The above is created for three vdW working fluids from a wetter fluid until a drier fluid ($C_{p}^{i}/R$ = 5.8, $C_{p}^{i}/R$ = 6.8, and $C_{p}^{i}/R$ = 7.6). Figure 3a shows the limit maximum efficiency obtained by the $\eta_{\mathrm{op}}$ and the efficiency obtained by the $W_{\mathrm{op}}$ functions depending on the condenser temperature. It can be seen that the difference between the efficiency obtained by the $\eta_{\mathrm{op}}$ and the $W_{\mathrm{op}}$ is small. Figure 3b shows the limit maximum net power output obtained by the $W_{\mathrm{op}}$ and the net power output obtained by the $\eta_{\mathrm{op}}$. In the same way as Δη, the $\Delta W_{n}$ is small.

## 4. Real Working Fluids

Figure 4a shows both analyzing optimal functions for the linear hydrocarbon series. The properties of the compounds are predicted using PC-SAFT, and the three required parameters are correlated as a function of the molecular weight [35] based on the parameters of the original version of PC-SAFT [32]. As expected, the predicted value of the optimal boiler temperature lies along the critical point of the hydrocarbon. The minimum values of the β-function and ω-function are fluids with $M_{w}$ = 49.42 and $M_{w}$ = 47.87, respectively. Therefore, in both optimization functions, the transition lies between propane and *n*-butane in the linear hydrocarbon series. Consequently, propane always performs as a wet fluid at optimal operating temperatures.

Additionally, over the functions calculated for linear hydrocarbons, Figure 4a displays non-linear hydrocarbons and refrigerants. It can be seen that the isobutane has a dry expansion range smaller than the *n*-butane with the same molecular weight for any condenser temperature. In the same way that the isobutane compared with the *n*-butane, the isopentane has a smaller range than the *n*-pentane. The differences are minor for hydrocarbons with a large molecular weight. The above concludes that the dry behavior depends on the number of atoms that compose the molecule, or, in the case of a molecular-based EOS, on the chain length, having a similar demeanor compared to other optimization functions [13,23]. The only fourth-generation refrigerant with a dry expansion in both cases is the R1233zd, similar to linear and branch hydrocarbons. In contrast, R1243zf and R1234ze only have a dry expansion by optimizing the net power output. The above means that these refrigerants never expand to a dry region for any value of the condenser temperature when the $\eta_{\mathrm{op}}$ is the optimizing function. R1234yf is not presented in Figure 4a because the expansion of the fluid is wet in its optimal conditions using both approaches and for any condenser temperature.

Figure 4b shows the value of the vapor fraction as a function of the condenser temperature obtained by the $\eta_{\mathrm{op}}$ and the $W_{\mathrm{op}}$ for R1234yf, R1243zf, and R1234ze. The continuous lines depict the vapor fraction obtained by the $\eta_{\mathrm{op}}$, and the segmented lines depict the vapor fraction obtained by the $W_{\mathrm{op}}$. A vapor fraction higher than 1.0 depicts a dry expansion. On the one hand, it can be seen that all the refrigerants have a wet expansion when $\eta_{\mathrm{op}}$ is used as the objective function. A vapor fraction higher than 0.86–0.89 is suitable for the expansion process in a turbine or expander [42]. Based on the above, the refrigerants R1234yf, R1243zf, and R1234ze can be used as working fluids for condenser temperatures between 320.00 K and 340.00 K when $\eta_{\mathrm{op}}$ is used as the optimizing function. Figure 4b exhibits that the value of the vapor fraction increases when the $W_{\mathrm{op}}$ is used as the objective function. However, the refrigerant R1234yf maintained a wet expansion and cannot be represented in Figure 4a. In the last cases, R1243zf and R1234ze have dry expansion when $W_{\mathrm{op}}$ is used. The range of the condenser temperature where the refrigerants can be used as a working fluid increases as its vapor fraction increases.

Figure 5 shows the boiler temperature values obtained by the $\eta_{\mathrm{op}}$ and the $W_{\mathrm{op}}$ as a function of the condenser temperature for each of the working fluids, as shown in Table 2. For each working fluid, two lines depict the value of the boiler temperature obtained by the $\eta_{\mathrm{op}}$ and the $W_{\mathrm{op}}$. For instance, the lines corresponding to the isobutane are rendered in black, as with the other hydrocarbons. The continuous line is the boiler temperature obtained by optimizing the limiting efficiency of the cycle. In contrast, the segmented line is the boiler temperature by calculating the optimal net power output. The gap between the boiler temperatures acquired through both approaches, $\Delta T_{3}$, is a near-constant value throughout the analyzed condenser temperature range. However, in the neighborhood of the critical point of the fluid, the gap evolves smaller, but it does not vanish. Figure 5 also displays a temperature range from 360.00 K up to 480.00 K, which matches the temperature range corresponding to the applications from low and moderate temperature sources, such as the geothermal, solar, and residual energy [9,40]. As indicated in Figure 5, each working fluid has a range of temperature depending on the condenser temperature, where the boiler’s operating temperature can move. However, this range varies slightly in wide ranges of temperature far from the critical point. Consequently, the range of operation is related directly to the value of the critical temperature of the working fluid. The above observation has been noted previously based on an exergetic analysis [15,20].

For example, the isothermal vertical line in Figure 5 is imposed at 293.15 K (20 °C) for the condenser. Figure 6 is built at this temperature, depicting the results for each working fluid. The blue and crimson bars depict the beginning of the expansion from the boiler using β- and ω functions, respectively. As expected, β-function provides higher temperature values than ω-function (see Figure 5). The temperature gaps, $\Delta T_{3}$, are highlighted in the green points at the secondary axis. Most of the hydrocarbons have a minor temperature range compared to the fourth-generation refrigerants. For instance, R1233zd has a $\Delta T_{3}$ = 6.17 K, and *n*-pentane has a $\Delta T_{3}$ = 5.22 K.

Additionally, the yellow bars represent the ranges of dry expansions. Solid yellow bars characterize the expansion of the β-function, while the hatched yellow bars render the expansion for the ω-function. Moreover, it can be seen that a fluid with a high critical temperature is not the driest. For instance, R1233zd has a smaller dry expansion range than the *n*-butane and a higher wet expansion than isobutane and *n*-butane for both approaches. Furthermore, three fourth-generation refrigerants have exclusively wet expansion for the two applied optimizing functions.

Additionally, the yellow bars represent the ranges of dry expansions. Solid yellow bars characterize the expansion of the β-function, while the hatched yellow bars render the expansion for the ω-function. Moreover, it can be seen that a fluid with a high critical temperature is not the driest. For instance, R1233zd has a smaller dry expansion range than the *n*-butane and a higher wet expansion than isobutane and *n*-butane for both approaches. Furthermore, three fourth-generation refrigerants have exclusively wet expansion for the two applied optimizing functions. The bottom of the yellow bars is the outlet of the turbine. In all cases, these temperatures are higher by optimizing the net power output and higher than the condenser temperature. The above can generate problems, such as overheating the condenser unit and higher exergy destruction [17]. Although R1233zd has a higher critical temperature than isobutane or *n*-butane, the value of the turbine outlet temperature is lower for both optimizing functions. Therefore, R1233zd is a working fluid more isentropic than isobutane and *n*-butane.

Unfortunately, the condensation in the turbomachinery is impossible to avoid at optimal conditions by optimizing the efficiency or net power output of the system. However, this handicap can be flanked by compromising performance and using the maximum point of saturation entropy as the turbine suction [22]. Figure 7 analyzes the decrease in performance of the ORC by comparing the efficiency of the system and the net power output of the cycle, using as reference the limiting efficiency and the maximum power output of the cycle determined by the β-function and ω-function, respectively. In this analysis, the condensator temperature is fixed at normal conditions ($T_{1}$ = 293.15 K), e.g., the temperature highlighted with the vertical crimson line in Figure 6.

Firstly, Figure 7a compares the decrease in the ORC efficiency caused for the expansion from the maximum point of saturation entropy with the efficiency obtained from both approaches. In this representation, $\Delta \eta_{1}$ compares the limiting maximum efficiency from the β-function with the cycle efficiency from the optimal net power output. As indicated, the loss in total efficiency is minimal. However, the change does not allow to avoid condensation. Further, to expand from the extremal point of the entropy saturation curve, $\Delta \eta_{2}$, ensure a dry expansion. Notwithstanding, the efficiency decrease is significant in some cases, particularly in fourth-generation refrigerants.

In contrast, Figure 7b resembles the decrease in the net power output of the system induced by the expansion from the maximum point of saturation entropy with the power obtained from both optimization functions. The analysis is similar to Figure 7a but optimizes the net power output. The energy decreases produced at the maximum efficiency, $\Delta W_{1}$, are similar in magnitude to the decreases produced by the expansion from the maximum point in the saturation curve, $\Delta W_{2}$. Both optimization approaches yield dramatically different operation conditions. For this reason, the selection of an optimization method is sensitive to other variables, mainly to the availability of the energy source. For this reason, in cases where the thermal energy source is stable, it is recommendable to obtain the maximum work of the system, compromising efficiency. Otherwise, total efficiency is critical. Therefore, a slight reduction in the produced energy is acceptable in the nominal working of the system.

The comparison of the different optimizing functions provides guidelines for selecting a recommendable working fluid from an engineering viewpoint, which is based on the temperature of an energy source independent of the mass flow of the source. For instance, the suitable temperature range for the heat transfer between the energy source and the working fluid through a heat exchanger lies from 5.0 to 8.0 °C [9,43]. Therefore, in the case of a constant temperature energy source of, for example, 115 °C, R1243zf is the recommendable working fluid for an ORC since the maximum net power is obtained at 105 °C. The above temperature is recommended as an initial temperature for the operation and testing of the performance of the cycle using the presented approach. In this example, R1243zf is the working fluid that obtains the operation conditions nearest to the maximum efficiency and maximum net power output that can reach any working fluid analyzed in this work. The latter analysis can be extended for any working fluid.

## 5. Conclusions

This work has been devoted to analyzing two objective functions for optimizing the optimal conditions of a single-stage ORC. It can be seen that both the ψ-function and ω-function obtained by the vdW EoS provide a good first look into the analysis of the expansion process of an ORC for several $C_{p}^{i}$ representing different working fluids. Some qualitative observations are that the value of the boiler temperature obtained by the $\eta_{\mathrm{op}}$ and the $W_{\mathrm{op}}$ increases for drier working fluids. The differences between both approaches are negligible at high values of the perfect gas heat capacity. The turbine outlet temperature obtained by $W_{\mathrm{op}}$ is higher than that obtained by the $\eta_{\mathrm{op}}$. However, for drier working fluids, this difference is negligible.

PC-SAFT EOS accurately represents the thermophysical properties of working fluids and their derivatives. PC-SAFT, both presented approaches, combined with the characterization of the maximum point in the saturation entropy envelope, are an excellent reference to obtain the optimal conditions for an ORC. The selection of optimal working fluid and operating conditions depends on many variables. This work provides guidelines and tools to help optimize the energy and variables of low-temperature power systems.

Applying the optimal net power output has engineering advantages compared to limiting fluid efficiency. On the one hand, among the benefits of maximizing the net power output of the system is the control of the condensation variables. On the other hand, depending on the availability of a heat source, efficiency is a less important variable than the supplied power. However, the generality of the limiting efficiency provides complete light to the analysis of a system. The comparison of the different optimization methods demonstrates that imposing a dry behavior of the system causes a dramatic difference in the obtaining efficiency but a similar demeanor in power output. From an engineering viewpoint, the selection of the optimization method must be completed under the mechanical, heat source, and utilization criteria.

## Figures and Tables

**Figure 1 entropy-25-00882-f001:**
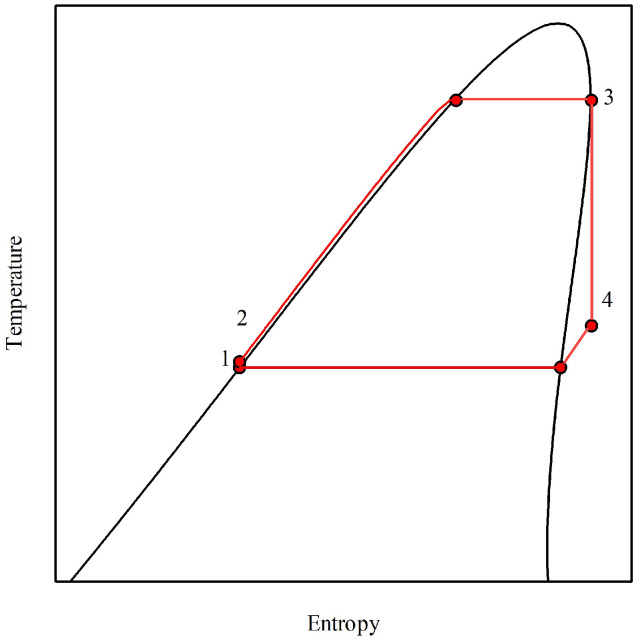
Schematic illustration of a single-stage ORC in the temperature vs. entropy projection for a near-isentropic working fluid (isobutane). (1-2) pump compression, (2-3) isobaric vaporization, (3-4) isentropic expansion at the turbine, and (4-1) isobaric condensation.

**Figure 2 entropy-25-00882-f002:**
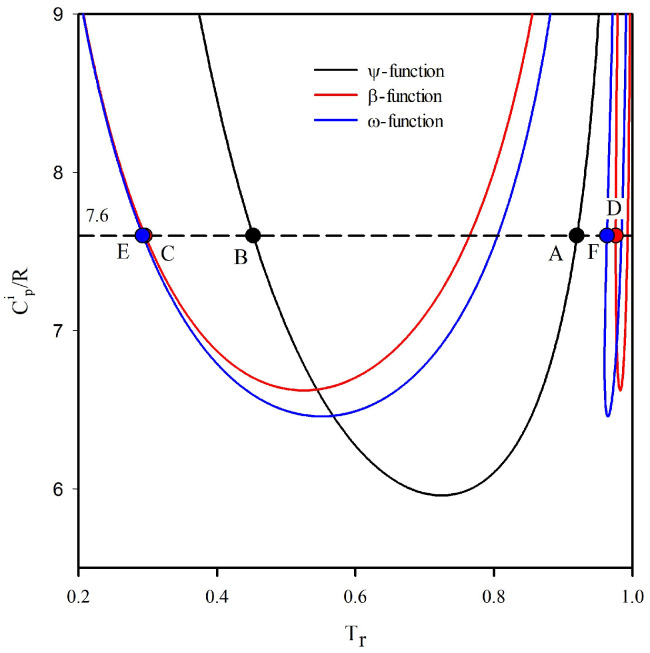
ψ-function in black, β-function in crimson, and ω-function in blue vs. heat capacities of the ideal gas, as predicted by the van der Waals EOS.

**Figure 3 entropy-25-00882-f003:**
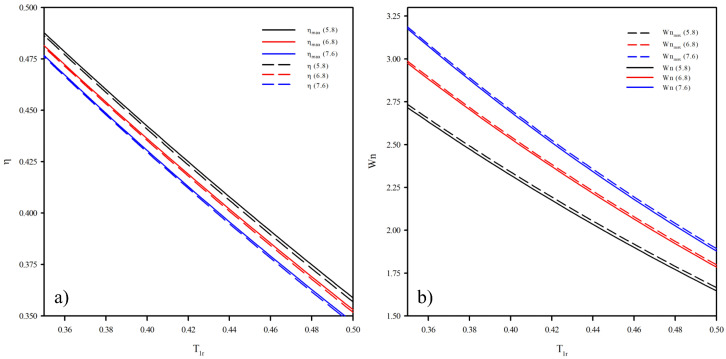
Comparison of the efficiency and the net power output obtained from Equation (Equation 7) in continuous lines and Equation (Equation 9) in segmented lines. (**a**) Efficiency from optimization functions. (**b**) Net power output from optimization function.

**Figure 4 entropy-25-00882-f004:**
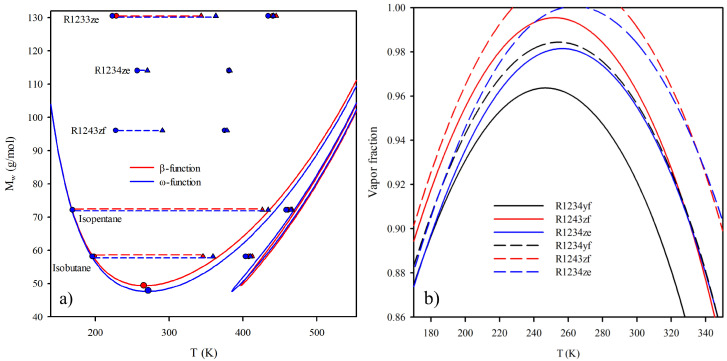
(**a**) Depiction of both β-function and ω-function of linear hydrocarbons series and other working fluids as a function of temperature. (**b**) Vapor fraction as a function of the condenser temperature obtained by the $\eta_{\mathrm{op}}$ and the $W_{\mathrm{op}}$.

**Figure 5 entropy-25-00882-f005:**
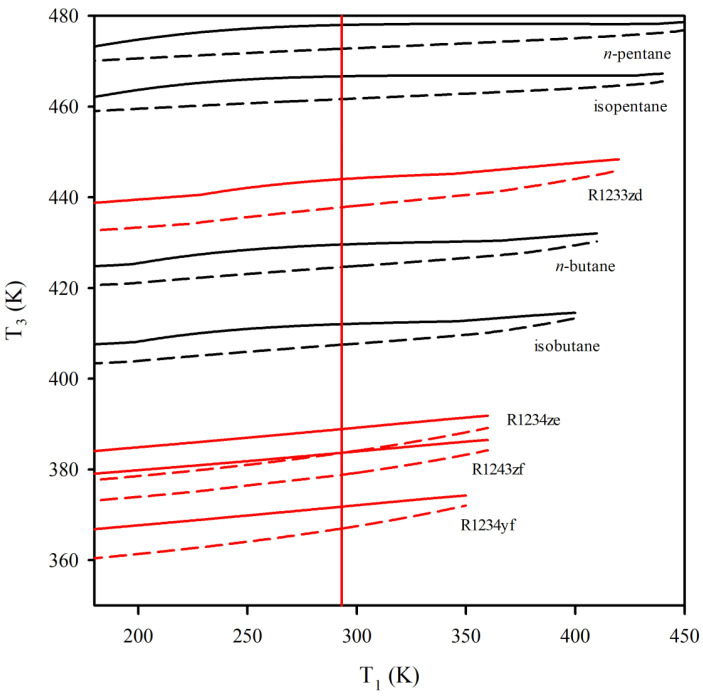
Value of the boiler temperature, $T_{3}$, obtained through the limiting optimal efficiency of the working fluid β-function in continuous lines and through the optimal net power output, $W_{\mathrm{op}}$, in segmented lines. Hydrocarbons are rendered in black, while fourth-generation refrigerants are depicted in crimson.

**Figure 6 entropy-25-00882-f006:**
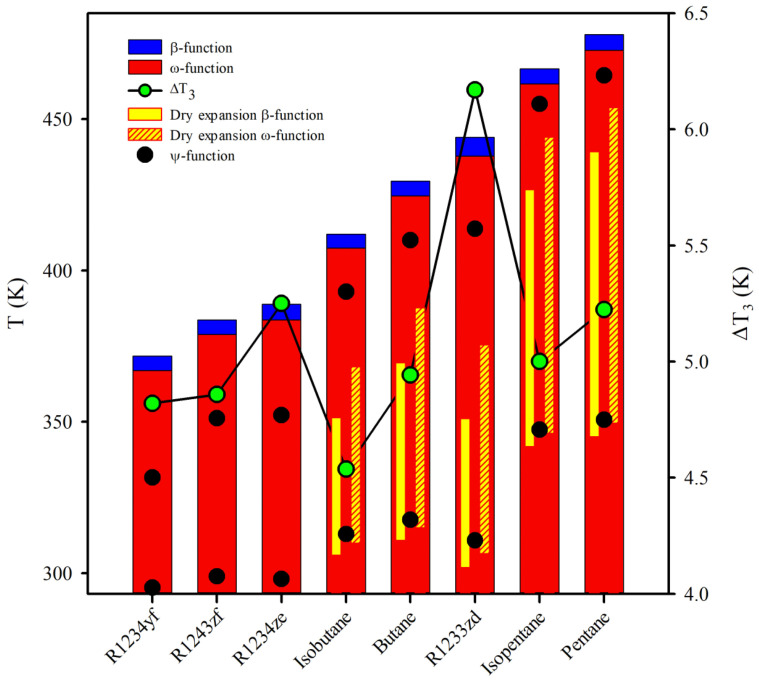
Behavior of the analyzed working fluids undergoing expansions, ensuring the limiting optimal efficiency and the maximum net power output of the system at 293.15 K.

**Figure 7 entropy-25-00882-f007:**
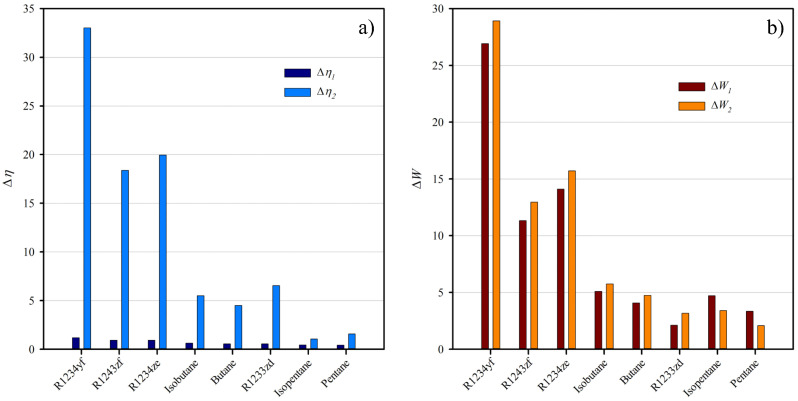
(**a**) Efficiency decrease, Δη, calculated as $\Delta \eta_{n}=100\left(\eta_{max}-\eta_{n}\right)/\eta_{max}$. (**b**) Net power output decrease, $\Delta W_{n}$, calculated as $W_{n}$ is $\Delta W_{n}=100\left(W_{max}-W_{n}\right)/W_{max}$. In both figures, the subscript *n* can be 1 or 2. Subscript 1 refers to the value obtained from ω-function and β-function for subfigures a and b, respectively. Subscript 2 refers to the expansion from the maximum point of the saturation entropy envelope.

**Table 1 entropy-25-00882-t001:** Parameters for the isobaric heat capacity of hydrocarbons and fourth-generation refrigerants used in Equation (5).

No.	Name	$\alpha_{0}$	$\alpha_{1}$	$\alpha_{2}$	$\alpha_{3}$	T. Range	Source
			$10^{2}$	$10^{5}$		$10^{-3}$ K	
1	isobutane	2.77420	3.593700	−1.1060	−43767.70	0.20–1.50	[22]
2	*n*-butane	2.76746	3.549044	−1.0783	−17384.86	0.20-1.50	[22]
3	isopentane	3.17210	4.461800	−1.3191	−60587.12	0.20–1.50	[22]
4	*n*-pentane	2.80526	4.500491	−1.3805	−9715.61	0.20–1.50	[22]
5	HFO-1234yf	−2.98468	7.149800	−8.1282	15699.73	0.22–0.37	[22]
6	HFO-1243zf	−3.08283	7.384900	−8.3955	16215.99	0.25–0.37	[22]
7	HFO-1234ze-E	−3.12707	7.490900	−8.5159	16448.70	0.18–0.38	[22]
8	HCFO-1233zd-E	−3.62707	7.990900	−9.0159	20448.70	0.20–0.45	[22]

**Table 2 entropy-25-00882-t002:** Molecular parameters for PC-SAFT of hydrocarbons and fourth-generation refrigerants.

No.	Name	R-Number	*m*	σ	$\varepsilon /k_{b}$	AARD	Ref.
				*Å*	K	${\tilde{v}}^{L}$	
1	isobutane	R600a	2.2012	3.7933	219.71	0.38	[22]
2	*n*-butane	R600	2.3316	3.7086	222.88	1.59	[32]
3	isopentane	R601a	2.5620	3.8296	230.75	1.53	[32]
4	*n*-pentane	R601	2.6896	3.7729	231.20	0.78	[32]
5	HFO-1234yf	R1234yf	2.8978	3.3648	174.91	1.70	[39]
6	HFO-1243zf	R1243zf	2.7112	3.3960	186.08	0.75	[40]
7	HFO-1234ze-E	R1234ze	3.2268	3.1909	173.87	0.60	[40]
8	HCFO-1233zd-E	R1233zd	3.1368	3.3909	202.51	2.84	[40]

## Data Availability

No new data were created or analyzed in this study. Data sharing is not applicable to this article.

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
