# Peer review of "Analysis of the Maximum Efficiency and the Maximum Net Power as Objective Functions for Organic Rankine Cycles Optimization"

_entropy, 2023, doi:10.3390/e25060882_

Round 1

Reviewer 1 Report

Overall, the paper is well-structured and well-written. Yet, it must be noted that the method is not adequately described, and the concise wording in some places - especially in chapter 3 - makes it quite challenging to comprehend. Therefore, improving the description of the methodology is recommended to enhance its interpretability.

Given that the primary objective of this paper is to examine the behavior of three functions (beta, omega, psi) across various equations of state and working fluids, the clarity of these function's physical content is essential, as, without them, any conclusions drawn from the functions will lack interest.

The current brief definitions between lines 127 and 135 fail to properly introduce these functions and their characteristics. Therefore, it is recommended to provide these equations in Chapter 3 or derive them in the appendix.

Other recommendations:

As the manuscript does not explicitly cover two-phase expansion, showcasing the ORC on a T-s diagram of a working fluid that avoids passing through the expansion process in the two-phase region may be beneficial (Figure 2.).

Is it possible that Wop in line 148 refers to power output, not efficiency?

If the source provides constant heat omega and psi must have the same Tr value at the optimum state. Have you investigated the correlation between the two functions?

Author Response

Dear Reviewer 1 
Please find the review in the attached file.
Regards

Reviewer 2 Report

The paper concerns the thermodynamics of recovery unit whose working fluid follows a sequence of transformations called Rankine cycle. Having the fluid an organic nature, the recovery technology can be named as "ORC".

The paper is well written, clear and presentation of overall text is more than satisfactory. However, the thermodynamics of ORC plants is well known, it fits inside books and lecture notes. Scientists know the difference of optimizing a cycle from the efficiency point of view as well as specific work and/or power recovery. These differences or approaches when a cycle is designed are known and a further effort from the Authors should be done in order to justify the originality, novelty, and, more generally, the overall interest of a new paper which treats a so-referenced matter. The availability of softwares which describe in a precise way the thermodynamic properties of a a numerous of fluids makes the problem easier and could avoid the use of complex or simplified equations of state. A paper improvement could be accomplished by adding in the paper: a) the engineering aspects which orient the design of a cycle to one direction or to another (efficiency, specific work, power); b) the influences (and they are) of the hot and cold sources to sustain a specific choice (when the maximization of the power produced, the flow rate of the working fluid is an important parameter and it influences the thermal power extracted by the hot source and that one reversed toward the cold fluid); c) the sizing of the components when a design choice is adopted. In this way the interest certainly increases and the paper "looses" a little bit the "training character" it has.

Author Response

Dear Reviewer 2
Please find the review in the attached file.
Regards

Round 2

Reviewer 1 Report

Comments were properly addressed, questions were answered in a satisfactory way.

Reviewer 2 Report

The Authors have answered to my remark. They modified the text of the previous version improving the previous version. This upgraded the paper which now can be published